# A Bibliometric Analysis of Melanoma Treated with Vaccinations Research from 2013 to 2023: A Comprehensive Review of the Literature

**DOI:** 10.3390/vaccines11061113

**Published:** 2023-06-19

**Authors:** Xinyu Wang, Qian-Nan Jia, Mengyin Wu, Mingjuan Liu, Jun Li

**Affiliations:** Department of Dermatology, State Key Laboratory of Complex Severe and Rare Diseases, Peking Union Medical College Hospital, Chinese Academy of Medical Sciences and Peking Union Medical College, National Clinical Research Center for Dermatologic and Immunologic Diseases, Beijing 100730, China; wangxinyu@student.pumc.edu.cn (X.W.); jqnnnsfw@126.com (Q.-N.J.); mengyin.wu@foxmail.com (M.W.); mingjuan.liu@mail.mcgill.ca (M.L.)

**Keywords:** melanoma, vaccine therapy, cancer vaccine

## Abstract

Backgrounds: Melanoma is a malignant tumor that originates from melanocytes and is known for its aggressive behavior and high metastatic potential. In recent years, vaccine therapy has emerged as a promising approach for the treatment of melanoma, offering targeted and individualized immunotherapy options. In this study, we conducted a bibliometric analysis to assess the global research trends and impact of publications related to melanoma and vaccine therapy. Methods: We retrieved relevant literature from the Web of Science database from the past decade (2013–2023) using keywords such as “melanoma”, “vaccine therapy”, and “cancer vaccines”. We used bibliometric indicators including publication trends, citation analysis, co-authorship analysis, and journal analysis to evaluate the research landscape of this field. Results: After screening, a total of 493 publications were included in the analysis. We found that melanoma and vaccine therapy have gained significant attention in the field of cancer immunotherapy, as evidenced by the numerous research output and increasing citation impact. The United States, China, and their organizations are the leading countries/institutes in terms of publication output, and collaborative research networks are prominent in this field. Clinical trials evaluating the safety and efficacy of vaccination treatment in melanoma patients are the focus of research. Conclusions: This study provide valuable insights into the novel research landscape of vaccine treatment of melanoma, which could inform future research directions and facilitate knowledge exchange among researchers in this field.

## 1. Introduction

Melanoma, the deadliest form of skin cancer, has been a growing concern worldwide due to its aggressive nature and increasing incidence rates. Traditional cancer therapies such as surgery, chemotherapy, and radiation therapy have shown limited therapeutic benefits in advanced melanoma cases, leading to an urgent demand for innovative approaches [1,2,3]. Therefore, there is a growing interest in immunotherapeutic approaches, including vaccines and vaccination, as promising strategies for melanoma treatment.

Vaccination, a proven method for preventing infectious diseases, involves stimulating the immune system to produce a response against specific antigens. The principle of cancer vaccines [4,5] is based on the concept of immunotherapy, which involves using the body’s immune system to target and eliminate cancer cells. In the context of melanoma [6,7,8], vaccines are designed to target antigens expressed by melanoma cells, prompting the immune system to mount an immune response against them. There are several types of cancer vaccines that have been explored for melanoma therapy, including peptide-based vaccines, DNA-based vaccines, viral vector-based vaccines, and whole-cell vaccines. For example, peptide-based vaccines [9] use peptides or proteins derived from melanoma cells as antigens. These antigens are then formulated into a vaccine that is administered to individuals at risk of developing melanoma or those who have already been diagnosed with melanoma. Furthermore, personalized vaccines tailored to the patient’s tumor antigens and neoantigen vaccines that target unique mutations in the tumor are also among the innovative approaches being studied in clinical trials to enhance the immune response against melanoma cells.

Bibliometric analysis [10], a quantitative approach to analyzing research publications, provides valuable insights into the research landscape, trends, and patterns in the melanoma treatment field. By analyzing bibliometric indicators, we can gain a comprehensive understanding of the research output and impact in the field of melanoma and vaccine/vaccination research. This analysis can help identify research hotspots, highlight influential authors and institutions, and provide directions for future research efforts.

## 2. Methods

We conducted a bibliometric analysis of melanoma and vaccine/vaccination research using the following steps:

### 2.1. Data Collection and Data Extraction

We searched relevant publications from Web of Science using appropriate keywords, such as “melanoma”, “vaccin*”, and “cancer vaccine”. Because WOS does not have Medical Subject Headings (MeSH), we used PubMed’s MeSH terms for the retrieval and the search formula is as follows #1: ((((((((((Immunotherapy, Active[MeSH Terms]) OR (Immunotherap*, Active[Title/Abstract])) OR (Vaccine Therap*[Title/Abstract])) OR (Vaccination[MeSH Terms])) OR (Vaccination*[Title/Abstract])) OR (Active Immunization*[Title/Abstract])) OR (vaccine*[Title/Abstract]))) OR (cancer vaccines[MeSH Terms])) OR (cancer vaccin*[Title/Abstract])), #2: ((((Melanoma[MeSH Terms]) OR (Melanoma*[Title/Abstract])) OR (Malignant Melanoma*[Title/Abstract]))), #3: #1 AND #2. The search was limited to articles published in English between 2013 and 2023 to capture the most recent research trends. Two of us screened the title and abstract independently. Only research focusing on melanoma and vaccines was included. All the cases included in this article were identified based on the International Classification of Diseases for Oncology (ICD/ICD-O) to ensure the accurate and comprehensive inclusion of related cases, which could help us to reduce clinical heterogeneity. Different opinions were adjudicated by the third party, and then the decision was made after discussions. We excluded studies that did not meet the inclusion criteria, which included the following (1). articles that did not focus on melanoma and vaccines; (2). reviews, systematic reviews, meta-analyses, correspondences, expert opinions, retracted articles, conference articles, or guidelines. For articles that met the criteria, we extracted data on bibliometric indicators, including publication trends (e.g., annual publication output), top research areas (e.g., keywords analysis, co-occurrence analysis), leading countries and institutions (e.g., country and institution rankings), prolific authors (e.g., author productivity, h-index), and citation analysis (e.g., citation counts, citation networks).

### 2.2. Data Analysis and Visualization

The TXT form data were imported into the VOSviewer1.6.18 software, which could visually analyze the keywords co-occurrence, authors, organizations, countries/regions, and reference citations, and co-citations. The visual elements commonly found in bibliometric analyses, such as circles, nodes, and lines, have their own unique meanings. Taking citation networks as an example, each circle represents a research paper or article. Each piece of literature is represented by a circle, and the size of the circle indicates its importance or the number of citations it has received. A larger circle suggests a highly influential or widely cited publication, while a smaller circle represents a less significant or less cited article. Connections or lines between circles can represent citation relationships between the literature. When one paper cites another, a line is drawn to connect the two circles. By visualizing these connections, we can observe the flow of knowledge and trace the influence of earlier works on subsequent research. In general, the thickness of the lines conveys the strength or frequency of citation relationships. If a paper is cited more frequently by other papers, the connecting line may appear thicker. Such lines can help us to identify influential literature or those with a higher number of citations in the research field. Colors are often used to categorize papers based on different criteria such as research themes, fields, or authors. By assigning different colors to specific groups, it becomes easier to identify and analyze clusters of related papers or distinct research areas within the bibliometric network. This visual representation helps in understanding the distribution and interconnectedness of literature across various domains.

## 3. Results

A total of 2063 publications related to melanoma and vaccine/vaccination were extracted from the Web of Science Core Collection database from 2013 to 2023. After literature screening by two researchers, 493 articles on the topic mentioned above were enrolled in this study. Figure 1 shows the details of the screening process.

### 3.1. Publication Trend

The number of publications of each year is displayed in Figure 2. Except for 2023 (publications in 2023 were incomplete), the number of relevant studies remained roughly stable at around 50 for all the years, with little variation. The year with the highest number of publications was 2018 (65 articles). The year with the lowest number was 2015 with 39 articles. The number of articles issued before 2018 showed a gradual increase. The apparent decrease from 2018 to 2019 may be due to the impact of the ongoing COVID-19 pandemic.

### 3.2. Journals of Publication

All the articles identified in this study were published in 178 journals. The number of publications varied from 1 to 34 in these journals. The journal with the most total citations is *Nature*. The top 10 journals comprise nearly 40 percent of the total publications, including *Cancer Immunology Immunotherapy*, *Oncoimmunology*, *Journal for Immunotherapy of Cancer* and *Frontiers in Immunology*. These journals also had comparatively higher impact factors, indicating the importance and quality of the research published in these journals.

### 3.3. Analysis of Countries/Regions, Organizations, and Authors

There were 42 countries/regions that reported on melanoma and vaccine/vaccination. The geographical distribution of publications is shown in Figure 3. The United States was the leading country in terms of publication output, with 157 publications (31.85% of total publications) originating from institutions in the United States. China ranked second with 94 publications (19.07%), followed by Germany with 18 publications (3.65%). Other countries with notable contributions included Italy (17, 3.44%), Argentina (16, 3.24%), South Korea (15, 3.04%), Japan (14, 2.84%), the Netherlands (14, 2.84%), Canada (12, 2.43%) and the United Kingdom (12, 2.43%).

Several leading institutions, including academic medical centers, cancer research institutes, and universities, were also identified. In total, 282 organizations had made contributions in this study area. The University of Virginia (USA) made the most contributions with 20 articles (40, 6.22%), followed by Radboud University Nijmegen (10, 2.03%). Of the top ten publishers (including ties, thirteen in total), four are from the United States, two from the Netherlands, six from China, and one from Switzerland. The details are shown in Figure 4.

The analysis of prolific authors in melanoma and vaccine/vaccination research identified 3364 researchers who have made significant contributions to the field. Among them, Craig Slingluff was the most prolific author with the largest publication number of articles (17, 3.45%). Followed by Walter Olson and Gina Petroni published 12 articles (2.43%). The details are shown in Figure 5.

### 3.4. Co-Occurrence Analysis of Keywords

The most frequently used keywords in the publications related to melanoma and vaccine/vaccination are shown in Figure 6. The size of the circle reflects the number of articles in which the term appears; the proximity of two associated terms indicates the relevance of the term based on the number of times they co-occur. The top keywords with the highest frequency were as follows: “melanoma” (n = 203), “immunotherapy” (n = 94), “vaccination” (n = 92), “dendritic cells” (n = 69), “cancer vaccines” (n = 60), “cancer immunotherapy” (n = 33), “nanoparticle vaccines” (n = 23); “adjuvants” (n = 22), indicating the central research themes in this field. Other frequently used keywords included “neoantigens”, “tumor microenvironment”, “personalized vaccination” and “immune checkpoint inhibitors”, reflecting the diverse aspects of research related to this topic.

### 3.5. Reference Citations and Co-Citations

Of the 439 papers, the analysis of reference citations discovered 145 nodes and 2195 links. The most cited article [11] was “An immunogenic personal neoantigen vaccine for patients with melanoma” by Ott, Patrick et al. in *Nature*, with a total of 1525 citations. The co-citation analysis was performed with 20 citations set as the minimum number of a cited reference. Of the 15001 cited references, 26 papers met the threshold with 281 links. In addition, the article entitled “Improved survival with ipilimumab in patients with metastatic melanoma” by Hodi Stephen in *The New England Journal of Medicine* ranked first with 53 co-citations [12]. The details are shown in Figure 7 and Figure 8.

## 4. Discussions

Over the years, there has been significant research and development in the field of melanoma and vaccine/vaccination. In recent years, several key trends and research directions have emerged, shaping the landscape of this field. Using thorough search criteria and rigorous reviews, we identified all the related documents, and systematically analyzed the basic characteristics of the literature, such as the journals, collaborative networks of countries, organizations and authors, co-occurrence of keywords, and citation network. The bibliometric analysis presented in our study provides valuable insights into the current state of research on melanoma and vaccine/vaccination, shedding light on the key research trends, challenges, and potential future directions.

While conventional, targeted, and combined therapies have significantly improved outcomes in melanoma patients, challenges remain, including the development of resistance mechanisms and limitations in certain patient populations [13,14,15]. Therefore, ongoing research is focused on identifying new targets and developing novel approaches to overcome these limitations. Vaccines and vaccination have revolutionized the field of medicine, playing a critical role in preventing infectious diseases. In recent years, there has been growing interest in harnessing the power of vaccines for cancer therapy [16].

The principle of cancer vaccines [17] is based on the concept of immunotherapy, which involves using the body’s immune system to target and eliminate cancer cells. Cancer cells often exhibit specific antigens on their surface that are different from normal cells. Cancer vaccines function by stimulating the immune system to recognize these antigens as foreign and mount an immune response against cancer cells. This can be achieved through various mechanisms, including the use of tumor-specific antigens, immune adjuvants, and delivery systems [18]. The treatment with vaccines/vaccination has many advantages, such as (1). specificity [19]: vaccines can be designed to target specific antigens or mutations that are unique to cancer cells, making them highly specific in their action. This allows for precise targeting of cancer cells while minimizing damage to normal healthy cells. (2). Personalization [20]: advances in genetic profiling and molecular diagnostics have enabled the development of personalized cancer vaccines that are tailored to an individual patient’s tumor profile. (3). Combination therapy [21]: vaccines can be used in combination with other cancer treatments, such as chemotherapy, radiation therapy, and immune checkpoint inhibitors, to enhance the overall therapeutic effect. This combination approach has shown promising results in clinical trials and has the potential to significantly improve patient outcomes. (4). Safety: vaccines are generally safe and well-tolerated, with fewer side effects compared to traditional cancer treatments. This makes them a promising option for patients who may not tolerate or respond well to other treatment modalities. Therefore, the use of vaccination in the treatment of melanoma has a very bright and exciting future. The goal of melanoma vaccination is to boost the body’s immune response against cancer cells, leading to tumor regression and improved outcomes. Clinical trials [22,23] have demonstrated that cancer vaccines can induce immune responses against melanoma antigens, resulting in tumor regression and improved survival outcomes in some patients.

The stable publication output from different countries and institutions indicates the sustained interest in advancing research on melanoma and vaccine/vaccination in the past ten years. The United States, China, and Europe are the leading contributors in terms of publication output and institutional productivity, with several renowned institutions in these regions leading the research efforts. The extensive investment in multiple regions reflects the heightened research efforts aimed at developing personalized vaccines as a promising immunotherapeutic approach for melanoma treatment. In addition, several research networks and clusters were identified, indicating active collaboration among researchers from different countries and institutions. As shown in Figure 2, the United States and China shared the strongest total link strength, indicating high-intensity cooperation between each other. In the bibliometric analysis of organizations, 282 organizations had made contributions in this study area. Most of the top-ranked institutions in terms of the number of articles published are from the United States, China, and Europe, which is essentially consistent with the epidemiology of melanoma. However, there is no single institution that is a clear leader in this area of research, which means that more in-depth studies and broader collaborations are needed to discover new breakthroughs. The analysis of prolific authors in this topic identified researchers who have made significant contributions to the field. Our study summarized a total of 3364 researchers participating in this research field. Many authors have published many articles, exhibited high productivity, and had a high index (namely a high impact factor, which can provide some indication of the visibility and influence of a journal). The prolific authors identified in this analysis represent the key contributors to the field, with their high productivity and impact on the research output.

The top research areas in melanoma and vaccine/vaccination research were identified through keyword analysis and co-occurrence analysis. Our analysis revealed that melanoma, immunotherapy, vaccination, dendritic cells [24], and cancer vaccines were the most frequently used keywords in publications, indicating the central research themes in this field. This is consistent with the current research focuses, which include the following: (1). attempts to find neoantigens or new adjuvants to improve the effectiveness of the treatment of melanoma [25]; (2). attempts to understand the immunological aspects of melanoma and developing effective vaccine/vaccination strategies [26].

From the keywords, we know that there are several types of cancer vaccines that have been explored for melanoma therapy, including peptide-based vaccines, dendritic cell-based vaccines, whole-cell vaccines, viral vector-based vaccines, and others. Peptide-based vaccines involve the use of short fragments of proteins that are derived from tumor-specific antigens. These peptides are selected based on their ability to bind to major histocompatibility complex (MHC) molecules, which are proteins responsible for presenting antigens to immune cells. When administered as a vaccine, these peptides can stimulate an immune response against melanoma cells by activating cytotoxic T cells to specifically recognize and attack melanoma cells that express the targeted antigens. DNA-based vaccines [27] involve the direct injection of DNA molecules encoding tumor-specific antigens into the body. Once inside cells, the DNA is taken up by the cells’ machinery, which then produces the antigens. These antigens are displayed on the cell surface, leading to the activation of immune cells and the induction of an immune response against melanoma cells expressing the targeted antigens. Viral vector-based vaccines [28] use harmless viruses, such as adenoviruses or lentiviruses, to deliver tumor-specific antigens into cells. These viruses are engineered to express the antigens, which are then displayed on the cell surface, leading to the activation of immune cells and the induction of an immune response against melanoma cells expressing the antigens. The vaccine platform employed can influence the magnitude and quality of the generated T cell response. The optimization of peptide selection, viral vector design, RNA stability and delivery, or DC maturation and antigen loading can enhance the intensity of T lymphocyte responses, ultimately contributing to improved therapeutic outcomes in melanoma patients. The intensity of the T lymphocyte response is a crucial aspect for evaluating the effectiveness of vaccine therapy, namely the potential for improved anti-tumor immune responses and better treatment outcomes in melanoma patients. We summarize the characteristics of different vaccine platforms in Table 1. In addition to those, there are several novel platforms-based vaccines, such as RNA-based vaccines, virus-like particle (VLP) vaccines (which mimic the structure of viruses without containing the viral genetic material), peptide-nanoparticle vaccines (controlled release of antigens and tunable immune responses), CRISPR-based vaccines (the flexibility to precisely engineer immune cells or tumor cells to express desired antigens, cytokines, or immune checkpoint inhibitors). These emerging vaccine platforms all provide exciting avenues for advancing melanoma immunotherapy. Taking RNA-based vaccines as an example, the concept behind RNA vaccines is to deliver synthetic RNA molecules encoding the neoantigens into the patient’s cells. Upon administration, RNA vaccines are taken up by antigen-presenting cells, such as dendritic cells. These DCs process the RNA and present the encoded neoantigens on major histocompatibility complex (MHC) molecules to T cells. The presentation of neoantigens to T cells leads to their activation, proliferation, and differentiation into effector T cells. CD8+ cytotoxic T cells recognize and directly target tumor cells expressing the encoded neoantigens, inducing tumor cell lysis. Concurrently, CD4+ helper T cells provide the necessary support by releasing cytokines, activating other immune cells, and enhancing the anti-tumor immune response. Additionally, RNA vaccines can induce immunological memory, providing long-term protection against tumor recurrence. This memory response enables a more rapid and robust immune response upon re-exposure to the same neoantigens, further aiding in tumor control. Several clinical trials have investigated the use of RNA vaccines encoding neoantigens as adjuvants in melanoma patients. Early-phase trials have shown encouraging results, with evidence of immune activation and tumor regression in some patients. In addition, clinical responses have been observed in some patients, including tumor regression and prolonged progression-free survival. The durable responses observed in certain individuals suggest the potential for long-term clinical benefits.

In addition, personalized vaccines [29] have emerged as a promising and interesting approach in the field of melanoma treatment. Melanoma is characterized by a high mutation rate, resulting in the expression of unique tumor-specific antigens. These neoantigens [30] can serve as targets for the immune system to recognize and attack melanoma cells. The process of personalized vaccine treatments for melanoma typically involves the following steps: (1). genetic analysis of the patient’s tumor: the tumor sample from the patient is analyzed through genetic sequencing to identify genetic mutations and alterations, including key genes that may drive the growth and spread of the tumor. (2). Vaccine design: based on the genetic information of the tumor, one or more vaccines containing specific antigens (such as mutated proteins or tumor-associated antigens) that are overexpressed or mutated in the patient’s tumor are designed and synthesized. (3). Vaccine preparation: the designed antigens are synthesized into a vaccine, and the dosage and administration route are adjusted based on the patient’s immune status and physical characteristics. 4. Vaccine administration: the synthesized personalized vaccine is injected into the patient’s body to activate their immune system and induce a specific immune response against the melanoma. Compared to traditional treatments, personalized vaccine treatment offers several advantages [31], which are as follows: (1). a high level of personalization: the vaccine can be tailored to the patient’s genetic information and tumor characteristics, better suiting their immune system, and potentially improving treatment efficacy. (2). Low toxicity: personalized vaccine treatment generally has lower side effects compared to traditional chemotherapy and radiation therapy, as it primarily works by activating the patient’s own immune system to fight the tumor, avoiding widespread damage to normal cells. (3). Potential for long-term immune memory: personalized vaccines have the potential to generate immune memory, which may help the patient’s immune system to recognize and target any recurrence of melanoma in the future. (4). Adaptability: personalized vaccines can be adapted and modified based on the patient’s response to treatment, melanoma characteristics, and evolving research findings. This flexibility allows for the ongoing optimization and customization of the treatment approach for each patient. Indeed, the off-the-shelf vaccines [32] that target common melanoma antigens cannot be ignored either for their broader applicability. The screening of off-the-shelf vaccines that can be readily available and cost-effective is also an active area of research, especially when their individualized nature might pose challenges in terms of manufacturing and scalability. The off-the-shelf vaccines [33] have the potential to provide a more feasible and accessible option for a larger population of melanoma patients with poor economic conditions.

Another keyword we focused on was adjuvants (or delivery systems). Recent research [34,35] has focused on developing novel adjuvants (delivery systems) that can enhance the immune response to cancer vaccines. For example, the use of Toll-like receptor agonists [36], cytokines, and other immune modulators [37] as adjuvants can stimulate the immune system and enhance the immunogenicity of cancer vaccines. In addition, the use of nanoparticles [38], liposomes, and other delivery systems can improve the stability, bioavailability, and targeting of cancer vaccines, leading to improved vaccine efficacy. We also paid attention to keywords about predictive biomarkers, such as tumor mutational burden [39] (TMB), microsatellite instability [40] (MSI), and immune gene expression profiles, which are being investigated as potential predictors of the response to cancer vaccines (biomarkers that can optimize patient selection and treatment outcomes)..

Another important keyword that should not be overlooked is combination therapy. The use of combination immunotherapies, including cancer vaccines in combination with other immune checkpoint inhibitors [41], adoptive cell therapies [42], or targeted therapies, has shown promising results in preclinical and clinical studies. For instance, checkpoint inhibitors are a type of immunotherapy that block proteins called immune checkpoints, which can inhibit the immune response against cancer cells. When combined with vaccines, checkpoint inhibitors can enhance the immune response generated by the vaccine, leading to improved clinical outcomes. Clinical trials combining personalized vaccines or other types of vaccines with checkpoint inhibitors, such as pembrolizumab [43] or nivolumab [44], have shown promising results in melanoma patients, with increased overall survival and durable responses observed in some cases. Similarly, adoptive cell therapies, such as chimeric antigen receptor (CAR) T-cell therapy [45] or tumor-infiltrating lymphocyte (TIL) therapy [46], involve the extraction, modification, and re-infusion of a patient’s own immune cells to specifically target and attack cancer cells. When combined with vaccines, adoptive cell therapies can enhance the efficacy of the immune cells by providing them with additional antigen targets to recognize and attack. Clinical trials [47] combining nano vaccines with adoptive cell therapies have shown promising results in breast cancer patients, with increased response rates and prolonged survival observed in some cases. Thus, these combination approaches may offer synergistic effects and provide new strategies for melanoma therapy.

Citation analysis is an important indicator of the impact and influence of publications in a particular field. The most highly cited publication related to melanoma and vaccine/vaccination is “An immunogenic personal neoantigen vaccine for patients with melanoma” Ott, Patrick et al. in *Nature*. This clinical trial evaluated the safety and efficacy of personalized vaccination in melanoma patients. They found that vaccination with neoantigens can both expand pre-existing neoantigen-specific T-cell populations and induce a broader repertoire of new T-cell specificities in cancer patients, tipping the intra-tumoral balance in favor of enhanced tumor control. The second most cited research [48] was a study published in *Nature* in 2016 that suggested that systemic RNA delivery to dendritic cells could exploit antiviral defense for melanoma immunotherapy. Another highly cited clinical study [49] published in *Science* in 2015 found that vaccination directed at tumor-encoded amino acid broadens the antigenic breadth and clonal diversity of antitumor immunity. Clinical trials play a crucial role in evaluating the effectiveness and safety of cancer vaccines, and the high citation count of such studies highlights their significance in advancing the field of melanoma immunotherapy. These high-quality studies represent the most cutting-edge research directions and are also the focus of most scholars’ attention. From the co-citation analysis, we found that a study published in *The New England Journal of Medicine* has the highest number of co-citations and total link strength. This article [12] found that ipilimumab, with or without a gp100 peptide vaccine, as compared with gp100 alone, improved overall survival in patients with previously treated metastatic melanoma, which confirmed the bright future of combination therapy with vaccination in advanced melanoma patients.

Despite the progress and growing interest in melanoma and vaccine/vaccination, several challenges exist. One of the challenges is the heterogeneity of melanoma tumors, which can vary in genetic mutations, immune microenvironment, and response to treatment. This heterogeneity poses difficulties in developing personalized vaccines that can effectively target and eliminate all tumor cells. Additionally, the variability of immune responses among patients poses challenges in predicting and optimizing vaccine responses in individual patients, as immune profiles can differ greatly from one patient to another. To address the issue of tumor heterogeneity, further research is needed to identify and validate melanoma antigens that are consistently expressed across different subtypes of melanoma tumors. This could involve utilizing advanced genomic and proteomic techniques to comprehensively characterize the antigenic landscape of melanoma tumors, and identifying novel antigens that can serve as effective targets for vaccines. Additionally, the development of strategies to target multiple antigens simultaneously, such as neoantigen combinations, could potentially enhance the efficacy of personalized vaccines by targeting a broader range of tumor cells. The issue of immune variability among patients could be addressed through the development of predictive biomarkers that can accurately predict the response to personalized vaccines in individual patients. This could involve identifying and validating biomarkers that are associated with vaccine response, such as immune cell profiling, gene expression signatures, or mutational burden. Integrating these biomarkers into clinical practice could help guide treatment decisions and optimize the selection of patients who are most likely to benefit from vaccination treatment.

Another challenge is the high costs associated with the development and administration of cancer vaccines. The complex and resource-intensive process of designing, manufacturing, and delivering personalized vaccines can be cost-prohibitive, limiting their accessibility and affordability for some patients and healthcare systems. Addressing these challenges will require further research efforts, technological advancements, and collaborations among researchers, clinicians, and policymakers. This could involve optimizing vaccine manufacturing processes to reduce costs, exploring alternative vaccine platforms or delivery methods that are more cost-effective, and advocating for reimbursement policies and funding support to ensure that vaccines are accessible to all eligible patients.

Additionally, exploring alternative adjuvants and vaccine delivery methods could potentially overcome the challenges of resistance and improve the efficacy of personalized vaccines. This could involve investigating the immunosuppressive mechanisms employed by tumors to evade immune responses, and developing combination therapies that target multiple points or pathways to enhance the anti-tumor immune response.

This analysis has some limitations. First, restrictions on article type (articles and letters) and language (English) may lead to inclusion bias. In addition, the absence of the latest research may also interfere with our understanding of the overall directions.

## 5. Future Recommendations and Conclusions

The future directions of vaccinations in melanoma are as follows: (1). neoantigen prediction: with the advent of genomic sequencing technologies, it has become possible to identify neoantigens from the tumor’s genetic mutations. Computational tools, such as artificial intelligence (AI), next-generation sequencing, bioinformatics, and machine learning algorithms, can analyze various factors, including mutation type, expression level, and HLA binding affinity, to select neoantigens that are highly immunogenic and specific to the melanoma. (2). Vaccine design and delivery strategies: vaccines can be designed using different vaccine platforms, including RNA, DNA, peptide, and dendritic cell-based vaccines. In addition to the vaccine platform, the incorporation of adjuvants or immune modulators can enhance the immune response triggered by vaccines, such as centralized and decentralized manufacturing models, cryopreservation of vaccine components, and streamlined logistics to ensure efficient and timely delivery to patients. (3). Combination therapies: clinical trials evaluating the efficacy of combination therapies are ongoing, and the results are eagerly awaited. New combinations or combinations of previous treatment modalities can be economically innovative. (4). Predictive biomarkers: identifying biomarkers that can accurately predict the response to cancer vaccines can help select patients who are most likely to benefit from the treatment and optimize treatment outcomes. (5). Clinical outcomes and long-term follow-up studies: the safety, efficacy, and long-term outcomes of melanoma vaccines and vaccination strategies need to be thoroughly evaluated in large-scale clinical trials to establish their effectiveness and safety profiles in diverse patient populations. Larger randomized controlled trials are needed to further evaluate the safety and efficacy of personalized vaccines in melanoma patients, including comparing their efficacy with standard treatments and assessing their long-term benefits. Here, we summarize some representative research on different research hotpots from major databases over the past two years in Table 2.

In conclusion, the field of melanoma and vaccine/vaccination has shown promising progress, but also faces challenges that need to be addressed for further advancements. The findings from this bibliometric analysis provide valuable insights into the current research landscape, trends, and challenges in this field. Further research directions can focus on identifying more targetable antigens, refining vaccine formulation and administration, studying biomarkers for patient selection, exploring combination therapies, and conducting long-term follow-up studies. Continued research in this field has the potential to optimize and improve the outcomes of vaccine treatment for melanoma patients.

## Figures and Tables

**Figure 1 vaccines-11-01113-f001:**
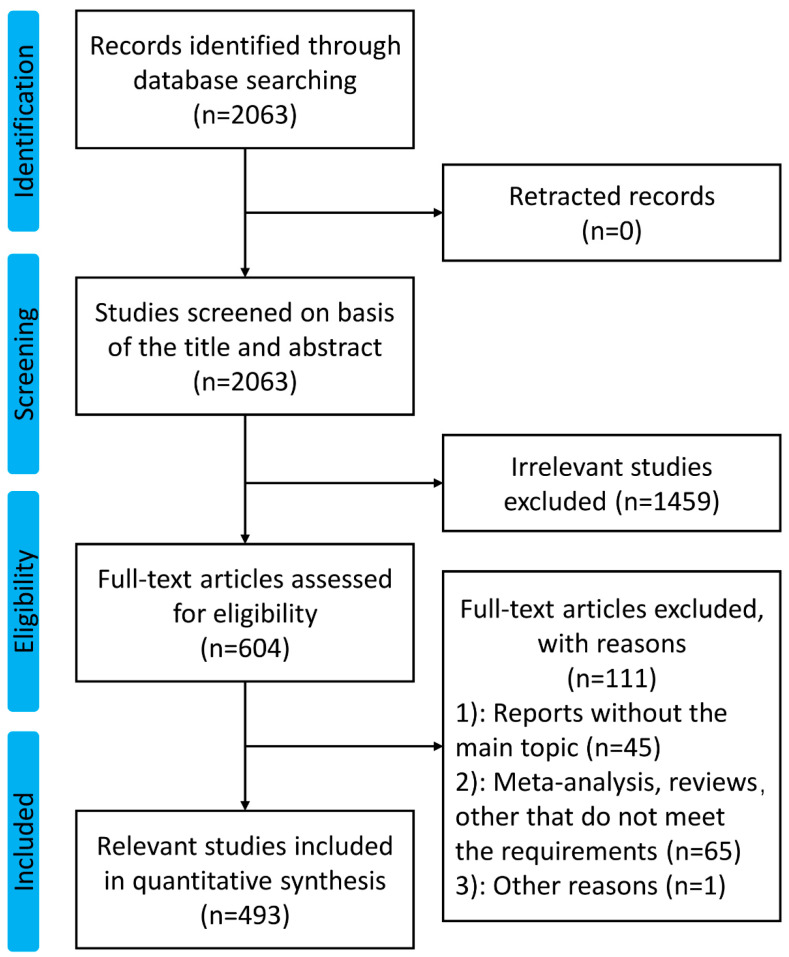
The flow diagram of this study.

**Figure 2 vaccines-11-01113-f002:**
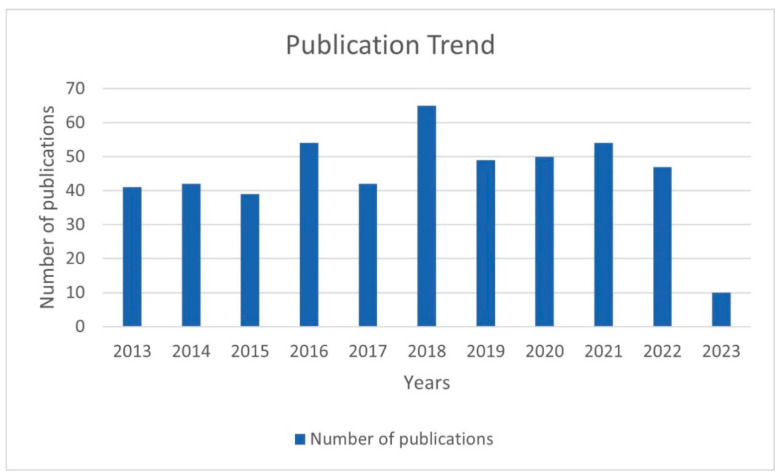
The number of publications of each year.

**Figure 3 vaccines-11-01113-f003:**
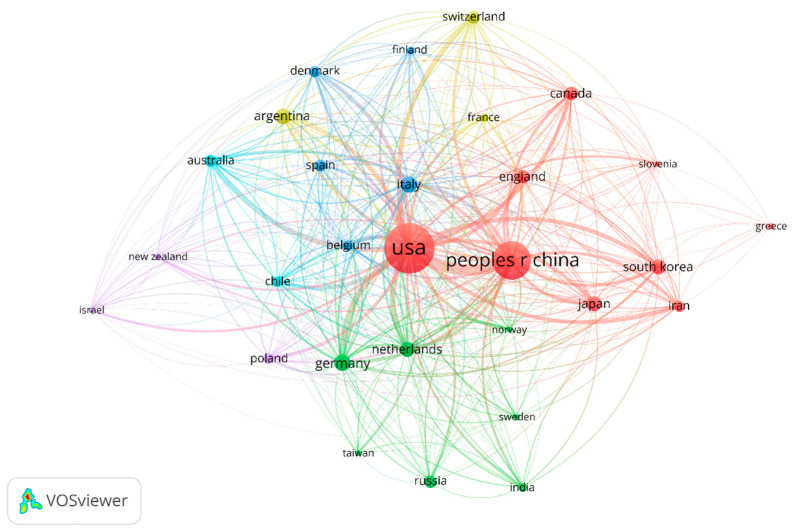
Geographical distribution of publications on melanoma and vaccination.

**Figure 4 vaccines-11-01113-f004:**
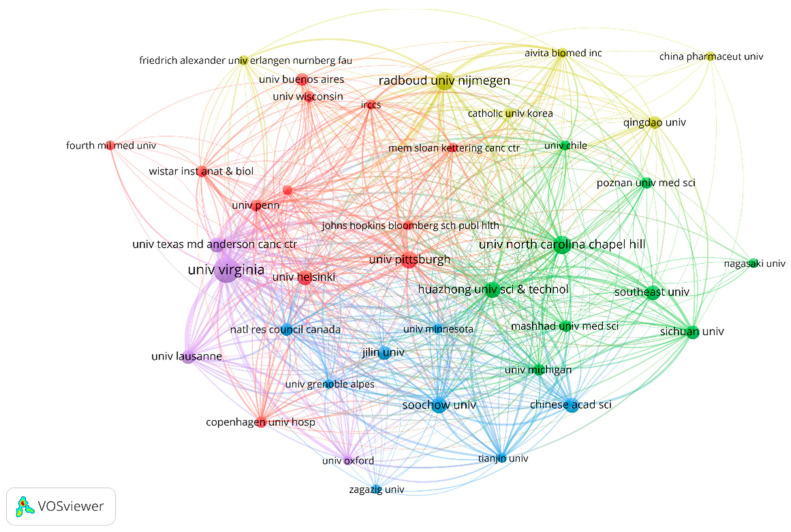
The collaborative network of organizations.

**Figure 5 vaccines-11-01113-f005:**
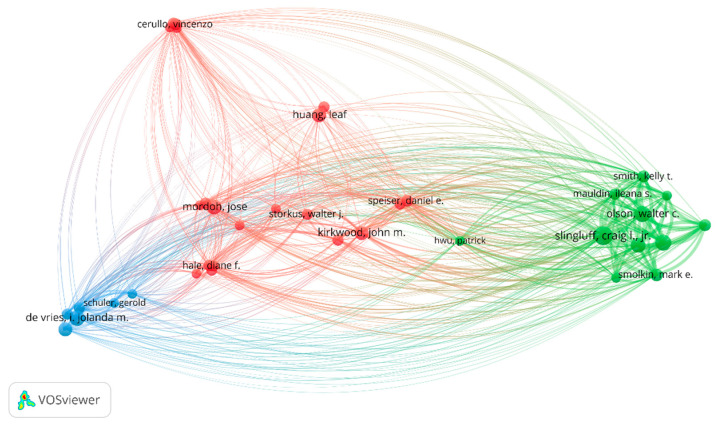
Leading authorship in melanoma and vaccination research.

**Figure 6 vaccines-11-01113-f006:**
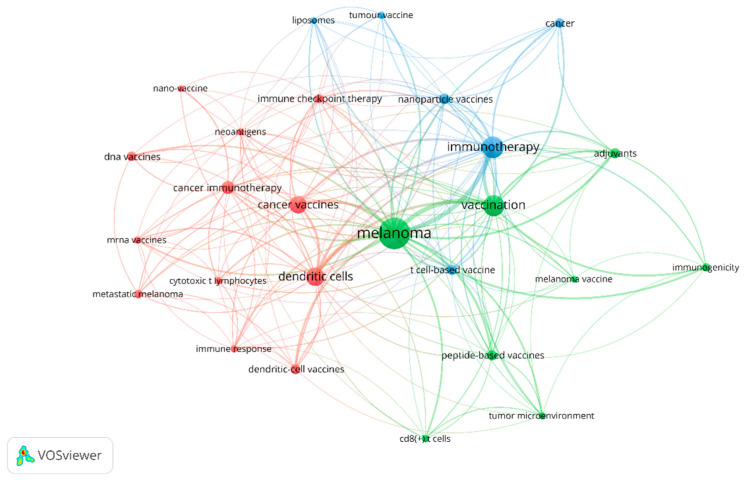
The co-occurrence network of keywords.

**Figure 7 vaccines-11-01113-f007:**
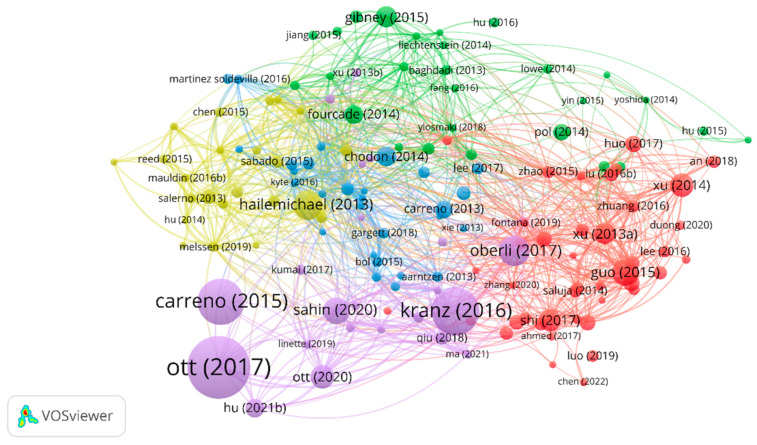
The analysis of reference citations.

**Figure 8 vaccines-11-01113-f008:**
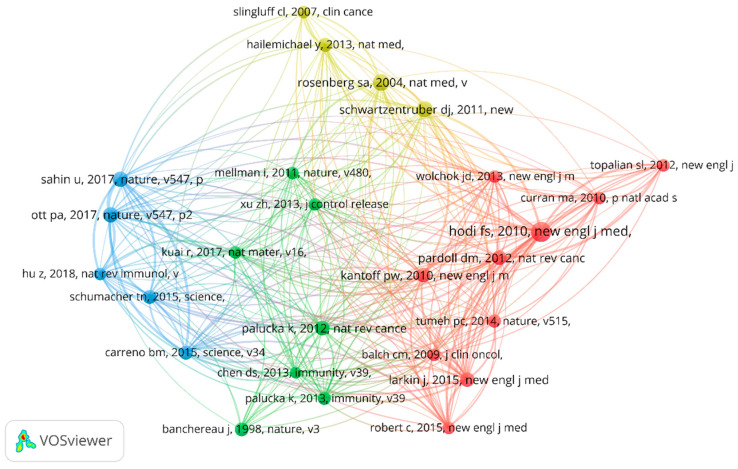
The analysis of reference co-citations.

**Table 1 vaccines-11-01113-t001:** Summary of different types of vaccine platforms for the treatment of melanoma.

Platforms	Characteristics	
**Peptide-Based Vaccines**	**Strengths**	1. Relatively safe and well-tolerated. 2. Have a well-defined antigenic target, can focus on specific tumor-associated antigens (TAAs). 3. Can be synthesized easily and at low cost. 4. Can induce both CD4+ and CD8+ T-cell responses.
	**Limitations**	1. Often exhibit limited immunogenicity, resulting in modest clinical efficacy. 2. Require patient-specific HLA matching, which can be challenging in large-scale applications. 3. May induce immune tolerance due to self-antigen presentation.
	**Targeted Population**	Suitable for patients with a well-characterized tumor antigen profile and a high likelihood of having HLA types matching the vaccine peptides.
	**Future Developmen** **ts**	1. Incorporate adjuvants or immune checkpoint inhibitors to enhance immunogenicity. 2. Explore personalized neoantigens to target tumor heterogeneity. 3. Investigate novel delivery systems to improve peptide stability and uptake.
**DNA-Based Vaccines**	**Strengths**	1. Can induce both humoral and cellular immune responses. 2. Have the potential for broad antigen coverage due to the expression of full-length TAAs. 3. Relatively easy to manufacture and can be modified rapidly. 4. Can be tailored to incorporate personalized neoantigens.
	**Limitations**	1. Low transfection efficiency limits the immune response generated. 2. The risk of integration into the host genome is a concern, although it is rare. 3. The need for electroporation or viral vectors for efficient delivery poses logistical challenges.
	**Targeted Population**	Have the potential to benefit a wide range of patients, especially those with solid tumors expressing well-characterized antigens and those who can receive efficient delivery methods.
	**Future Developmen** **ts**	1. Improve transfection efficiency through advanced delivery techniques. 2. Enhance antigen expression levels to optimize immune responses. 3. Combine DNA vaccines with other immunotherapies to enhance clinical outcomes.
**Viral Vector-Based Vaccines**	**Strengths**	1. Can efficiently deliver TAAs to target cells, enhancing antigen presentation. 2. Have the potential to induce robust immune responses. 3. Can be modified to express multiple antigens simultaneously. 4. Can amplify the vaccine effect through viral replication.
	**Limitations**	1. Safety concerns. 2. Pre-existing immunity to the vector may limit vaccine effectiveness. 3. Immune evasion mechanisms employed by the virus may hinder vaccine efficacy.
	**Targeted Population**	Can benefit patients with advanced melanoma who have intact immune systems and limited pre-existing immunity against the vector.
	**Future Developmen** **ts**	1. Improve vector design to enhance tumor targeting and antigen expression. 2. Overcome pre-existing immunity through the development of novel vectors. 3. Investigate combination therapies for synergistic effects.
**Dendritic Cell-Based Vaccines**	**Strengths**	1. Offer the possibility of targeting multiple antigens simultaneously. 2. Can be modified ex vivo to enhance their immunogenicity. 3. Can be personalized based on patient-specific antigens.
	**Limitations**	1. The complex and expensive process of dendritic cell isolation and activation limits scalability. 2. The limited lifespan of dendritic cells after infusion hinders long-term immune responses. 3. The immunosuppressive tumor microenvironment may hinder dendritic cell function.
	**Targeted Population**	May benefit patients with advanced melanoma who have intact immune systems and accessible tumor tissues for dendritic cell isolation.
	**Future Developmen** **ts**	1. Explore artificial antigen-presenting cells (aAPCs) as a substitute to streamline the process. 2. Use of genetic engineering to enhance their antigen presentation and T-cell activation capabilities. 3. Investigate combination therapies to overcome immune suppression.
**Whole Cell-Based Vaccines**	**Strengths**	1. Can provide a broad range of antigens. 2. Can induce both humoral and cellular immune responses. 3. Can address tumor heterogeneity and immune escape mechanisms. 4. The process of whole cell-based vaccine production is relatively straightforward.
	**Limitations**	1. Quality control and standardization can be challenging. 2. The immune response may not be specific to tumor antigens. 3. The potential for immune tolerance to self-antigens exists.
	**Targeted Population**	May benefit patients with advanced melanoma, particularly those with a high tumor mutational burden or tumor heterogeneity.
	**Future Developmen** **ts**	1. Identify optimal antigen combinations to maximize immune responses. 2. Incorporate other immunotherapies to enhance vaccine efficacy. 3. Develop personalized vaccine formulations based on individual tumor neoantigen profiles.

**Table 2 vaccines-11-01113-t002:** Summary of recent advances associated with different related hotpots from major databases.

Titles	Year	Journals	Databases	Topics
1. Characterization of the T cell receptor repertoire and melanoma tumor microenvironment upon combined treatment with ipilimumab and hTERT vaccination.	2022	*J Transl Med*	PubMed	Tumor microenvironment Cancer vaccine
2. Phase I/II clinical trial of a helper peptide vaccine plus PD-1 blockade in PD-1 antibody-naïve and PD-1 antibody-experienced patients with melanoma (MEL64).	2022	*J Immunother Cancer*	PubMed	PD-1 Cancer vaccine
3. Dendritic cell vaccines targeting tumor blood vessel antigens in combination with dasatinib induce therapeutic immune responses in patients with checkpoint-refractory advanced melanoma.	2021	*J Immunother Cancer*	PubMed	Dendritic cells Neoplasm antigens
4. Melanoma stem cell vaccine induces effective tumor immunity against melanoma.	2023	*Hum Vaccin Immunother*	Embase	CD8-Positive T-cell Cancer vaccine
5. Targeting the tumor microenvironment by liposomal Epacadostat in combination with liposomal gp100 vaccine.	2023	*Sci Rep*	Embase	Tissue distribution Tumor microenvironment
6. A randomized controlled trial of long NY-ESO-1 peptide-pulsed autologous dendritic cells with or without alpha-galactosylceramide in high-risk melanoma.	2023	*Cancer Immunol Immunother*	Cochrane	Dendritic cell NY-ESO-1
7. An update of cutaneous melanoma patients treated in adjuvancy with the allogeneic melanoma vaccine vaccimel and presentation of a selected case report with in-transit metastases.	2022	*Front Immunol*	Cochrane	Immunologic adjuvants Cancer vaccine
8. Engineered antibody cytokine chimera synergizes with DNA-launched nanoparticle vaccines to potentiate melanoma suppression.	2023	*Front Immunol*	Cochrane	DNA vaccines Nanoparticles
9. Liposomal celecoxib combined with dendritic cell therapy enhances antitumor efficacy in melanoma.	2023	*J Control Release*	WOS	Combination therapy Dendritic Cells

## Data Availability

Data available upon request from the authors.

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
