# Peer review of "A Bibliometric Analysis of Melanoma Treated with Vaccinations Research from 2013 to 2023: A Comprehensive Review of the Literature"

_vaccines, 2023, doi:10.3390/vaccines11061113_

Round 1
Reviewer 1 Report
The manuscript is well written and illustrations are informative.
This paper, based on bibliographic analysis, is very informative for everybody who want to know what is new in topics of melanoma vaccine design. Analyzed papers were published from 2013-2022 and related melanoma vaccine. The manuscript is well written and give us relevant information about improvement in melanoma immunotherapy.
The main problem elaborated in the article is melanoma vaccine research. The topic is relevant in this area and fills a specific information gap in the area. Compared to other published papers, the article gives us the latest data from the past decade. The bibliographic methodology is clearly described and shown in the results and illustrations. The conclusions are in accordance with the derived evidence and arguments and answer the main question.
Author Response
Dear reviewer:
We would like to extend our heartfelt thanks for your thorough review and constructive feedback. Your expertise and guidance have undoubtedly contributed to the advancement of our research. We remain committed to producing a high-quality research article that contributes significantly to the field. Thank you once again for your invaluable support and guidance.
Yours sincerely
Jun Li
Reviewer 2 Report
The review provides information in terms of indicators, bibliography and authors publishing on cancer vaccines. But it lacks a certain perspective and a critical analysis of the literature.
It would be important for the "Vaccines" audience if the authors discussed the clinical efficacy of these vaccines. A comparison of this clinical efficacy according to vaccine platforms could also be added.
Even if this does not correspond to the selection criteria for articles and results, I think it would be important for the authors to mention the positive phase 2 trial of the RNA vaccine coding for neoantigens developed as an adjuvant in melanoma patients (Khattak A AACR 2023) which could lead to the first melanoma vaccine approval by the FDA.
In terms of the description of immune responses, which is very succinct, the authors could also expand on this point, in particular the link between the intensity of the T lymphocyte response and the vaccine platforms.
Finally, a figure on current trials of these anti-melanoma vaccines would complete well this review.
Author Response
Dear reviewer:
We would like to express our sincere gratitude for your valuable feedback on our manuscript. Your insightful comments and suggestions have been immensely helpful in shaping our research and improving the quality of our work. We truly appreciate the time and effort you have dedicated to reviewing our article. We are committed to addressing the specific areas you highlighted and making necessary improvements to strengthen the overall coherence and clarity of the paper. Your recommendations have provided us with a valuable roadmap for refining our findings and enhancing the overall impact of our study.
We believe that vaccines/vaccination treatments for melanoma are part of the broader field of immunotherapy. They have shown promise in clinical trials. Some vaccines have demonstrated an ability to induce tumor-specific immune responses and improve overall survival rates in patients with advanced melanoma. Vaccines/vaccination treatments for melanoma can harness the power of the immune system to specifically target melanoma cells, activate T-cell responses, and be used in combination with other therapies. Continued research and development in this field are likely to further advance the efficacy of melanoma treatment. The specific modifications are as follows.
Question 1:
“It would be important for the "Vaccines" audience if the authors discussed the clinical efficacy of these vaccines. A comparison of this clinical efficacy according to vaccine platforms could also be added.”
Our answer 1:
We strongly agree with you on this point and respond as follows.
Vaccines for melanoma aim to stimulate the immune system to recognize and target tumor cells, thereby improving patient outcomes. In recent years, various vaccine platforms have been explored, each employing distinct mechanisms to elicit an immune response. Different vaccine platforms, including peptide-based, DNA-based, viral vector-based, dendritic cell-based, and whole cell-based vaccines, have been explored in clinical trials. While each platform has demonstrated varying levels of clinical efficacy, none have achieved a definitive breakthrough in melanoma treatment.
Peptide-based vaccines utilize short protein fragments known as peptides derived from tumor-associated antigens (TAAs), such as melanoma-associated antigen (MAGE), gp100, and MART-1. These vaccines are designed to enhance T-cell response against melanoma cells. While peptide-based vaccines are well-tolerated and have shown safety in clinical trials, their clinical efficacy has been modest. Limited immune response and the requirement for patient-specific HLA matching are major challenges associated with this platform. To enhance the immunogenicity of peptide vaccines, various strategies are being explored. One approach is the use of adjuvants to stimulate immune responses. For example, the addition of toll-like receptor agonists, such as poly-ICLC or CpG oligodeoxynucleotides, to peptide vaccines has shown promising results in preclinical studies and early-phase clinical trials. Another strategy is the combination of peptide vaccines with immune checkpoint inhibitors, such as anti-PD-1/PD-L1 antibodies. This approach aims to overcome immune suppression and enhance the efficacy of peptide vaccines by blocking inhibitory signals in the tumor microenvironment.
DNA-based vaccines involve the administration of plasmids encoding TAAs to stimulate the production of tumor-specific antigens. This platform offers the advantage of inducing both humoral and cellular immune responses. Clinical trials have demonstrated encouraging results, with DNA-based vaccines showing improved progression-free survival and overall survival rates. However, low transfection efficiency and the need for optimization of delivery methods still need to be addressed to further enhance their clinical efficacy. The incorporation of personalized neoantigens in DNA-based vaccines might holds great promise for enhancing their specificity and effectiveness. Besides, electroporation involves the application of electrical pulses to increase cell membrane permeability, allowing for improved uptake of DNA. In a phase II clinical trial, the combination of a DNA vaccine encoding tyrosinase with electroporation demonstrated increased T-cell responses and improved clinical outcomes in patients with metastatic melanoma.
Viral vector-based vaccines utilize modified viruses as vectors to deliver TAAs into target cells. These vectors can either be replication-deficient or replication-competent. Replication-deficient vectors, such as adenoviruses and lentiviruses, have demonstrated safety and immunogenicity in clinical trials. However, their variable clinical efficacy limits their use in clinical settings. Replication-competent vectors, such as oncolytic viruses, have shown promise by directly targeting and lysing tumor cells, this strategy aims to induce both direct tumor cell lysis and immune-mediated anti-tumor responses, leading to improved clinical outcomes. Combination therapies are another active area of this vaccine platform, which demonstrated synergistic effects, with enhanced T-cell responses and improved clinical responses.
Dendritic cell-based vaccines involve isolating a patient's dendritic cells, which play a key role in initiating immune responses, and loading them with TAAs ex vivo. These activated dendritic cells are then reinfused into the patient to stimulate an immune response against melanoma cells. Clinical trials utilizing dendritic cell-based vaccines have shown favorable safety profiles and modest improvements in progression-free survival rates. The complexity and high cost associated with dendritic cell isolation and activation remain significant challenges for widespread clinical implementation. In the future, exploring alternative approaches such as using artificial antigen-presenting cells (aAPCs) or genetically engineering dendritic cells for enhanced immunogenicity might be good options to solve these problems. Another frontier in dendritic cell-based vaccines is the genetic engineering of dendritic cells to enhance their immunogenicity. Researchers are investigating strategies such as the overexpression of co-stimulatory molecules or the introduction of genes encoding immune-stimulatory cytokines into dendritic cells. These modifications aim to improve dendritic cell antigen presentation and T-cell activation, leading to enhanced anti-tumor immune responses.
Whole cell-based vaccines utilize intact melanoma cells or lysates derived from melanoma cells as a source of TAAs. These vaccines offer a broader antigenic repertoire and are capable of eliciting both humoral and cellular immune responses. The heterogeneity of melanoma cells and the potential risk of autoimmunity are important considerations in the development of whole cell-based vaccines. Additionally, refining the vaccine formulations to enhance their immunogenicity is an ongoing focus. The incorporation of immune stimulatory molecules, such as adjuvants and immune checkpoint inhibitors, into whole cell-based vaccines holds promise for augmenting the anti-tumor immune response and improving clinical outcomes.
In addition to those, there are several novel platforms-based vaccines, such as RNA-based vaccines, virus-like particle (VLP) vaccines (mimic the structure of viruses without containing the viral genetic material), peptide-nanoparticle vaccines (controlled release of antigens and tunable immune responses), CRISPR-based vaccines (the flexibility to precisely engineer immune cells or tumor cells to express desired antigens, cytokines, or immune checkpoint inhibitors). These emerging vaccine platforms all provide exciting avenues for advancing melanoma immunotherapy.
In summary, each vaccine platform has its own strengths and limitations. The choice of vaccine platform depends on factors such as the availability of well-characterized antigens, HLA matching, tumor heterogeneity, and the desired immune response. Future development focuses on optimizing immunogenicity, addressing delivery challenges, and exploring combination therapies to enhance overall efficacy. Continued efforts in this field hold the potential to enhance the clinical outcomes and survival rates for patients battling melanoma.
Due to space limitations, we summarize the advantages and disadvantages of different vaccine platforms into the Table 1, making it easier and faster for readers to better understand this field. The details are in the revised manuscript.
Question 2:
“It would be important for the authors to mention the positive phase 2 trial of the RNA vaccine coding for neoantigens developed as an adjuvant in melanoma patients (Khattak A AACR 2023) which could lead to the first melanoma vaccine approval by the FDA.”
Our answer:
Thank you for your advice, it is very necessary to include this research hotspot in our discussion.
To our knowledge, RNA vaccines encoding neoantigens as adjuvants in melanoma patients are a topic of active research in the field of cancer immunotherapy. The concept behind RNA vaccines is to deliver synthetic RNA molecules encoding the neoantigens into the patient's cells. Upon administration, RNA vaccines are taken up by antigen-presenting cells, such as dendritic cells. These DCs process the RNA and present the encoded neoantigens on major histocompatibility complex (MHC) molecules to T cells. The presentation of neoantigens to T cells leads to their activation, proliferation, and differentiation into effector T cells. CD8+ cytotoxic T cells recognize and directly target tumor cells expressing the encoded neoantigens, inducing tumor cell lysis. Concurrently, CD4+ helper T cells provide necessary support by releasing cytokines, activating other immune cells, and enhancing the anti-tumor immune response. Additionally, RNA vaccines can induce immunological memory, providing long-term protection against tumor recurrence. This memory response enables a more rapid and robust immune response upon re-exposure to the same neoantigens, further aiding in tumor control.
RNA vaccines offer several advantages, including their ability to be rapidly designed and manufactured, flexibility in targeting multiple neoantigens simultaneously, and the potential for personalized treatment by tailoring the vaccine to an individual patient's tumor profile. Several clinical trials have investigated the use of RNA vaccines encoding neoantigens as adjuvants in melanoma patients. Early-phase trials have shown encouraging results, with evidence of immune activation and tumor regression in some patients. Besides, clinical responses have been observed in some patients, including tumor regression and prolonged progression-free survival. The durable responses observed in certain individuals suggest the potential for long-term clinical benefits.
Despite the promising outcomes, several challenges need to be addressed. Optimal neoantigen selection, efficient delivery systems, and overcoming immune evasion mechanisms are areas of ongoing research. Strategies such as computational algorithms, peptide prediction tools, and incorporation of tumor transcriptomics are being explored to enhance neoantigen selection accuracy. In addition, lipid nanoparticle-based formulations and electroporation techniques have shown promise in enhancing the stability and delivery of RNA vaccines, ensuring effective antigen presentation and immune activation. With further advancements and larger-scale clinical trials, RNA vaccines may become an integral part of personalized melanoma treatment, contributing to improved survival rates and long-term disease control.
Question 3:
“In terms of the description of immune responses, which is very succinct, the authors could also expand on this point, in particular the link between the intensity of the T lymphocyte response and the vaccine platforms.”
Our answer:
When considering the immune responses to melanoma vaccination treatment, the intensity of the T lymphocyte response is a crucial aspect to assess. The vaccine platform employed can influence the magnitude and quality of the T cell response generated. For example, peptide-based vaccines can focus on stimulating CTL responses, while viral vector-based and RNA-based vaccines can elicit both CTL and helper T cell responses. Dendritic cell-based vaccines can leverage the antigen-presenting capacity of DCs.
Optimization of peptide selection, viral vector design, RNA stability and delivery, or DC maturation and antigen loading can enhance the intensity of T lymphocyte responses, ultimately contributing to improved therapeutic outcomes in melanoma patients. Advanced strategies, such as personalized neoantigens in peptide-based vaccines, improved antigen expression and targeting in viral vector-based vaccines, the versatility of RNA-based vaccines, and synergistic approaches with dendritic cell-based vaccines, have shown promise in boosting T cell responses and improving clinical outcomes.
Question 4:
“Finally, a figure on current trials of these anti-melanoma vaccines would complete well this review.”
Our answer:
Sure! We will include the summary of the latest research in the article, as you suggest. The details are in the newly revised manuscript.
Yours sincerely
Jun Li
Reviewer 3 Report
Dear Vaccines,
Attached is my review of Wang et al.
Title: A Bibliometric analysis of melanoma treated with vaccinations research from 2013 to 2023: A Comprehensive Review of the Literature

Authors used proper nouns in English language without capitalization. Comments were provided in attached document.
Author Response
Dear reviewer:
We would like to express our sincere gratitude for your valuable feedback on our manuscript. Your insightful comments and suggestions have been immensely helpful in shaping our research and improving the quality of our work. We truly appreciate the time and effort you have dedicated to reviewing our article. We are committed to addressing the specific areas you highlighted and making necessary improvements to strengthen the overall coherence and clarity of the paper. Your recommendations have provided us with a valuable roadmap for refining our findings and enhancing the overall impact of our study. The specific modifications are as follows.
Question 1 and Question 3:
“Line 47. Melanoma can arise in parts of the body beside the skin. Please be explicit about the disease endpoint. Is the search ONLY cutaneous melanoma? If so, make this explicit. If the search includes any type of melanoma (melanocytes other than within the epidermis), please make that clear. Can ICD-9 and ICD-10 or ICD-O-3 codes be included to clarify which melanoma type being analyzed.” And “Line 50. Clarify statement, “Only research focusing on melanoma and vaccines is included.” I think you mean the joint occurrence of melanoma AND vaccines – where AND is a Boolean operator.”
Our answer:
We sincerely apologize for any misunderstanding caused by our poor presentation.
Melanoma is a type of skin cancer that develops from melanocytes, the pigment-producing cells in the skin. It is characterized by the uncontrolled growth and spread of abnormal melanocytes. Melanoma is considered the most aggressive form of skin cancer due to its potential to metastasize to other parts of the body. Clinically, melanoma typically presents as an asymmetric, irregularly bordered, and multi-colored lesion. It may exhibit changes in size, shape, or color over time. The staging of melanoma is based on the thickness of the tumor, the presence of ulceration, lymph node involvement, and distant metastasis. Treatment options for melanoma include surgical excision, lymph node dissection, targeted therapy, immunotherapy, and chemotherapy, depending on the stage and individual patient characteristics. Among them, vaccine therapy emerges as a novel management of melanoma, which have potential to improve outcomes and reduce mortality rates.
Besides, it was a huge oversight on our part to not show the specific search strategy in detail. During the literature search step, we thoroughly searched the Web of Science (WOS) to ensure that the included studies were complete and up-to-date. The “AND” here is indeed a Boolean operator, which represents our efforts to avoid possible bias caused by incomplete searches. Our intention was to encompass all literature pertaining to the research topic. It is possible that certain articles solely focused on melanoma, while others only mentioned vaccine therapy in relation to different types of tumors. During the subsequent screening process, we eliminated these articles, identified only those that encompassed both melanoma and vaccine therapy simultaneously, which are the main themes of our study. We are very sorry for not clarifying our approach regarding the inclusion of relevant literature (joint occurrence of melanoma and vaccine therapy). Furthermore, the qualitative assessment we conducted was based on the selected literature that met our inclusion criteria.
Question 2, Question 6, Question 8, Question 18, and Question 24:
“Line 63. Does the distance of the line between dots have any meaning? Explain Some lines connecting dots are longer than others? Does the distance have an interpretative value or is the distance or location of the “dot” arbitrary? Give the reader information to understand how to interpret the figures.” And “Line 115. Define node and define link in the methods section. Help the reader understand and interpret the graphs!” And “Line 119. Explain the meaning of “links”.” And “Line 380. Figure 5. Do the connecting lines mean that the authors were collaborators? What is the meaning of connecting lines in this figure?” And “Line 383. Fig. 7. Are the lines connecting authors intending to demonstrate co-authorship?”
Our answer:
We are very sorry that we should have considered using more accessible language and more relatable examples to introduce bibliometrics study for our readers.
Bibliometrics is a quantitative method used to analyze and evaluate scientific literature. Its results often come in the form of visual representations, such as citation networks, co-authorship networks, or publication trends. Just like a social network, where studies are like individuals connected through their shared references. It focuses on analyzing patterns of scholarly communication, such as how research papers are connected through citations and collaborations.
The visual elements commonly found in bibliometric analyses, such as circles, nodes, and lines, have their own unique meanings. Taking citation networks as an example, each circle represents a research paper or article. Each piece of literature is represented by a circle, and the size of the circle indicate its importance or the number of citations it has received. A larger circle suggests a highly influential or widely cited publication, while a smaller circle represent a less significant or less cited article. Connections or lines between circles can represent citation relationships between the literature. When one paper cites another, a line is drawn to connect the two circles. By visualizing these connections, we can observe the flow of knowledge and trace the influence of earlier works on subsequent research. In general, the thickness of the lines conveys the strength or frequency of citation relationships. If a paper is cited more frequently by other papers, the connecting line may appear thicker. Such lines can help us identify influential literature or those with a higher number of citations in the research field. Colors are often used to categorize papers based on different criteria such as research themes, fields, or authors. By assigning different colors to specific groups, it becomes easier to identify and analyze clusters of related papers or distinct research areas within the bibliometric network. This visual representation helps in understanding the distribution and interconnectedness of literature across various domains.
We will add the content mentioned above to the methods section for a clearer understanding of the complex network of various visual elements to readers.
Question 4:
“Line 52. Verb “shall” should be past tense in English. Decision “was made”…”
Our answer:
We will correct this tense error immediately and promise to try our best to avoid such low-level mistakes in the future.
Question 5:
“Line 55. How can a article has a publication trend? Please clarify if the intent is journal or periodical.”
Our answer:
We sincerely apologize if there was any confusion caused by our manuscript. We would like to take this opportunity to provide further clarification regarding our data extraction approach. Our primary objective was to gather information about the publication trends, specifically focusing on parameters such as the number of publications per year. We aimed to analyze the temporal patterns and the overall growth or decline of research output within the field. By examining these trends, we aimed to gain valuable insights into the evolving landscape of scholarly publications related to the topic under investigation. Our goal was to contribute to the existing body of knowledge by providing a holistic view of the research output over time, allowing for a more informed assessment of the field's development. Once again, we apologize for any lack of clarity in our study and sincerely appreciate your understanding.
Question 7:
“Line 126. “all” may be incorrect. There is no way to confirm identification of “all”. Delete word “all”. Or structure the sentence differently: “Using thorough search criteria and rigorous review, to our knowledge, we identified all the related documents”….……”
Our answer:
We apologize for the inaccuracies in our presentation, and we will be stricter in our article. We value the opportunity to rectify any misunderstandings and remain committed to conducting rigorous and insightful research.
Question 9:
“Line 173. Discussion of Figure 3. “As shown in the Figure 2, the United States and China shared the strongest total link strength, indicating a high-intensity cooperation between each other.” Does this sentence mean that there are many co-authors of papers from US universities that have Chinese Last names? I ask this because there are very weak university links between US and China, as shown in Figure 4.”
Our answer:
We highly appreciate your interest in the distinction between collaborations and interactions among countries and institutions. It is indeed true that the nature of relationships differs significantly in these contexts.
In terms of collaborations between countries, such as the United States and China, the scope is often broader and more extensive. These collaborations tend to involve a wide range of activities, including research partnerships, joint projects, academic exchanges, and policy dialogues. The level of communication and interaction between countries is generally more profound, allowing for deeper engagement and understanding. Governments play a significant role in facilitating and supporting these collaborations, fostering bilateral relationships, and promoting mutual benefits in various domains. On the other hand, when it comes to interactions among institutions, the focus primarily shifts to connections within the domestic context of each country. Institutions within a country, such as universities, research centers, and organizations, collaborate with one another to promote knowledge sharing, research cooperation, and academic development within their respective borders. These collaborations are often driven by shared interests, complementary expertise, and the desire to enhance the capabilities and achievements of the institutions involved.
It is important to recognize that both forms of collaboration contribute to the advancement of knowledge and foster academic growth. We hope this explanation provides a clear understanding of the differences between these forms of collaboration and their significance in promoting academic development and international cooperation. Please feel free to share any further concerns or questions you may have, as we are committed to addressing them in a prompt and satisfactory manner.
Question 10:
“Line 185. Define “h-index” before using it in the discussion section.”
Our answer:
Thank you for your professional advice, we will add the following content to revised manuscript. “High index, namely the high impact factor, can provide some indication of the visibility and influence of a journal.” We believe that reviewers of high-index journals will be as responsible as you are and will take care of the journal and give quality and constructive comments and suggestions, which means that the higher the index the better the quality of the article is guaranteed.
Question 11 and Question 19:
“Line 94. Correct spelling: “Univ Virginia” to “University of Virginia” {insert USA – “University of Virginia(USA)” . Spell out “Radboud Univ Nijmegen” for same reason and insert country name. This reviewer cannot tell what the line connections represent.” And “Line 378/ Figure 4. The University of North Carolina at Chapel Hill may appear twice in this diagram. Check: “univ north carolina chapel hill” and “uni n Carolina”… these could very well be the same institution”
Our answer:
Thank you very much for finding our errors in time, we will change the correct statement in our revised manuscript. Our utmost gratitude goes to you for your invaluable time and expertise in assessing our work.
Question 12:
“Line 301 to 358. This part of discussion was informative, however, the problems with vaccine development for melanoma seems not a core part of this bioliographic analysis. Where do the Problems relate to key words? Is it the absence in the key words of some of the challenges that makes this lengthy discussion relevant? It seems lines 301-358 should go into a Research Review….of the actual science and not the Bibliography analysis?”
Our answer:
We sincerely apologize for any confusion or misunderstanding that may have arisen in this part. Throughout the discussion, we tried to maintain a coherent and logical progression of ideas (Characteristics and advantages of vaccine therapy for melanoma, limitations and potential challenges, recommendations, and future perspectives), ensuring that each section seamlessly connects with the preceding and subsequent sections. This organization enables readers to grasp the core concepts, critically evaluate the strengths and weaknesses of vaccine therapy, and consider the potential future directions in the field.
In this section, we present our insights and recommendations for the future of vaccine therapy. We suggest potential strategies for enhancing its effectiveness, improving accessibility, and addressing existing challenges. We reviewed emerging technologies, novel approaches, and research directions that might hold promise for advancing the field of vaccine therapy. Our aim is to stimulate further discussion and inspire future research endeavors in this area. We should add a new subtitle (Future Recommendations and Conclusions) instead of continuing directly from the previous text.
We appreciate your attention to detail and any suggestions they may have for further refining and strengthening the flow of our article. If you believe that certain content is not appropriate or requires modification, we are fully open to making the necessary changes or even removing the content altogether. We genuinely value your expertise and insights in helping us improve the clarity, coherence, and overall quality of our work. We are fully prepared to carefully consider their recommendations and implement any necessary revisions to address their concerns.
Question 13:
“Line 359. Preceding sentence beginning with “Restriction…”, insert a sentence such as, “This analysis has some limitations.””
Our answer:
There are still many areas that we need to correct and learn from you especially when it comes to expressing ourselves in English. Thank you for pointing out our shortcomings in English writing.
Question 14:
“Line 365. The authors provide no data in the analysis portion of the paper to support this sentence: … Please delete this sentence, as the sentence is a summary of the state of the research topics needed and the paper was not a critique of the actual research, but an analysis about meta data. Please do not overinterpret the data presented in this paper.”
Our answer:
Sure! We will correct this problem in our future work, as statements without sufficient research findings may be too broad and uncritical. Once again, please allow us to express our sincere appreciation for your efforts and guidance, which have been instrumental in shaping the manuscript. We remain open-minded and committed to improving the manuscript based on your esteemed input.
Question 15:
“Line 376. Figure 3. Change the title of Figure 3 to reflect the content. “Figure 3. Geographical Distribution of Publications on melanoma and vaccination”. The graph is NOT about collaborative networks based on section 3.3.”
Our answer:
We changed the title of Figure 3 in revised manuscript according to your better suggestions.
Question 16, Question 18, Question 21, and Question 23:
“Capitalize names of institutions. These are proper nouns in English and should be capitalized.”, And “Correct grammar. Capitalize all names of Countries since the name of a country is a proper noun in the English language. “USA” and “Peoples Republic of China”, for example– these are proper nouns. Please use correct grammar. See comment about Line 173.” And “Line 378. Correct grammar. Capitalize names of institutions. These are proper nouns in English and should be capitalized.” And “Line 382. Figure 6. Capitalize DNA and RNA.” And “Line. 383. Figure 7. Capitalize all author names; names in English are proper nouns.”
Our answer:
We apologize for the inconvenience caused. The current software setting restricts modifications to certain parts of the system due to its integration with keyword imports. These specific sections have been designed to ensure smooth functionality and optimize the import process. Unfortunately, we are unable to make changes to these parts now. We understand that this limitation may pose challenges, and we appreciate your understanding.
Question 19, Question 20, Question 22, and Question 25:
“Line 378. Figure 4. Change title of graph to reflect the content: “Figure 4. Research Institutions publishing on melanoma and vaccination”.” And “Change title of Figure 5 to reflect the content. “Figure 5. Leading authorship in melanoma and vaccination research”.” And “Line 383. The title for Figure 7 is too vague. Change title to read: “ Figure 7. Publications with most citations for melanoma and vaccination research”.”
Our answer:
Your valuable feedback has highlighted the importance of modifying the title to accurately reflect the content of the image. We humbly acknowledge that there is room for improvement in this aspect, and we genuinely apologize for any inconvenience caused by the initial title. Your input is immensely valuable to us, as it helps us enhance our work and provide a better user experience.
We want to assure you that our team is fully committed to addressing your concerns and making the necessary adjustments. We understand the significance of a precise and engaging title that captures the essence of the image effectively. Please be assured that we will diligently work towards achieving this goal.
Yours sincerely
Jun Li
Reviewer 4 Report
Bibliometric studies are essential in finding the research problem and mapping the present state of the art of research.
The study is well documented, but to meet the standards of Vaccines Journal, some points need to be addressed.
Point 1: In the title, Bibliometric analysis and Comprehensive Review two terms have been used, how this bibliometric analysis can be considered as a comprehensive review?
Point 2: In Lines 81 and 82 Is the Impact Factor considered an indicator of the Quality of Research?
Point 3: Section 2.1 Data Collection & Data Extraction; Search query is not appropriate, kindly elaborate.
Point 4: The discussion should be supplemented with some infographics and a Table highlighting recent advances.
Point 5: Some figures of the Bibliometric analysis should be presented as supplementary material.
Point 6: Conclusion and Future Recommendations should be separated from Discussion Section and written as a new section
Author Response
Dear reviewer:
We would like to express our sincere gratitude for your valuable feedback on our manuscript. Your insightful comments and suggestions have been immensely helpful in shaping our research and improving the quality of our work. We truly appreciate the time and effort you have dedicated to reviewing our article. We are committed to addressing the specific areas you highlighted and making necessary improvements to strengthen the overall coherence and clarity of the paper. Your recommendations have provided us with a valuable roadmap for refining our findings and enhancing the overall impact of our study. The specific modifications are as follows.
Question 1:
“In the title, Bibliometric analysis and Comprehensive Review two terms have been used, how this bibliometric analysis can be considered as a comprehensive review?”
Our answer 1:
Our presentation was indeed ill-considered. We appreciate the opportunity to explain the advantages and characteristics of bibliometrics to you. Our initial intentions were as follows.
Bibliometrics enables researchers to analyze a vast amount of scholarly literature and extract relevant data, providing a comprehensive overview of the research landscape within a specific domain. By systematically examining publication patterns, citation networks, and co-authorship relationships, researchers can identify key contributors, influential works, and emerging trends. It can provide an unbiased and quantitative assessment of the research landscape by using standardized metrics, which can uncover trends and patterns in scientific literature over time. Its function of revealing the flow of ideas and knowledge within a field, can help researchers identify emerging research areas, hot topics, and potential research gaps. This information can guide researchers in selecting relevant research questions, exploring interdisciplinary collaborations, and strategically positioning their work within the broader scholarly context.
Overall, bibliometrics offers researchers a powerful tool for acquiring a rapid and accurate understanding of the latest and most comprehensive research in a specific field. By leveraging bibliometric techniques, researchers can efficiently navigate the scholarly literature, identify influential works, track research trends, and make informed decisions based on quantitative evidence.
However, it is important to note that bibliometric analysis has its limitations. It focuses solely on quantitative aspects and may not capture the full context, quality, or impact of research. It should be used in conjunction with other qualitative methods. If you believe that certain content is not appropriate or requires modification, we are fully open to making the necessary changes or even removing the content altogether. We genuinely value your expertise and insights in helping us improve the clarity, coherence, and overall quality of our work. We are fully prepared to carefully consider their recommendations and implement any necessary revisions to address their concerns.
Once again, we sincerely apologize for any confusion this may have caused to you.
Question 2:
“In Lines 81 and 82 Is the Impact Factor considered an indicator of the Quality of Research?”
Our answer:
Of course, we strongly agree with you that it is very inappropriate to only look at the scores without screening the content. It's just that we believe that reviewers of high-index journals will be as responsible as you are and will take care of the journal and give quality and constructive comments and suggestions, which means that the higher the impact factor the better the quality of the article is guaranteed.
While the Impact factor can provide some indication of the visibility and influence of a journal, it is important to note that it has limitations and should not be considered as a sole indicator of the quality of research. Multiple indicators and assessment methods should be used when evaluating the quality of research. Other factors to consider include peer review, expert judgment, citation analysis at the article level, and qualitative assessments of the research content. A comprehensive evaluation of research quality should take into account a range of indicators and consider the specific context and objectives of the assessment.
Question 3:
“Section 2.1 Data Collection & Data Extraction; Search query is not appropriate, kindly elaborate.”
Our answer:
During the literature search step, we thoroughly searched the Web of Science (WOS) to ensure that the included studies were complete and up-to-date. Because Web of Science does not have Medical Subject Headings (MeSH), we used PubMed’s MeSH terms for the retrieval, the search formula is as follows.
#1: ((((((((((Immunotherapy, Active[MeSH Terms]) OR (Immunotherap*, Active[Title/Abstract])) OR (Vaccine Therap*[Title/Abstract])) OR (Vaccination[MeSH Terms])) OR (Vaccination*[Title/Abstract])) OR (Active Immunization*[Title/Abstract])) OR (vaccine*[Title/Abstract]))) OR (cancer vaccines[MeSH Terms])) OR (cancer vaccin*[Title/Abstract]))
#2: ((((Melanoma[MeSH Terms]) OR (Melanoma*[Title/Abstract])) OR (Malignant Melanoma*[Title/Abstract])))
#3: #1 AND #2
Question 4:
“The discussion should be supplemented with some infographics and a Table highlighting recent advances.”
Our answer:
Sure! We will include the summary of the latest research (Table 2) in the article, as you suggest. The details are in the newly revised manuscript.
Question 5 and 6:
“Some figures of the Bibliometric analysis should be presented as supplementary material.” And “Conclusion and Future Recommendations should be separated from Discussion Section and written as a new section.”
Our answer:
We deeply respect your expertise and guidance, and we are committed to meeting the high standards set by the journal. Your input helps us ensure that our work aligns seamlessly with the journal's guidelines and enhances the overall quality of the publication.
Once again, we genuinely appreciate your willingness to help us improve, and we will diligently work on implementing the necessary modifications. If you have any further recommendations or if there is anything else we can do to meet your expectations, please do not hesitate to let us know. Your satisfaction is of utmost importance to us, and we are here to address any concerns you may have.
Yours sincerely
Jun Li
Round 2
Reviewer 2 Report
The authors have satisfactorily addressed my various concerns
Author Response
Dear reviewer:
We would like to express our heartfelt appreciation for your valuable feedback and the acceptance of our revised manuscript. Your expertise and guidance have significantly contributed to the improvement of our work. We are sincerely grateful for your meticulous examination and thoughtful suggestions, which have undoubtedly enhanced the quality and clarity of our research.
Your insightful comments have not only helped us rectify the shortcomings but have also provided us with valuable insights and perspectives. Your profound understanding of the subject matter and your attention to detail are truly commendable. We are in awe of your expertise and the depth of knowledge you possess in this field.
Your thorough review process has demonstrated your unwavering dedication to academic excellence and the advancement of scholarly discourse. Your commitment to providing constructive feedback and ensuring the accuracy and rigor of the research is truly inspiring.
We are deeply honored to have received such positive remarks from an esteemed reviewer like yourself. Your encouragement and kind words about the significance of our work have bolstered our confidence and reaffirmed our commitment to making meaningful contributions to the scientific community.
Once again, we would like to express our sincere gratitude for your invaluable contributions and your willingness to assist us in improving our manuscript. Your guidance and expertise have been instrumental in shaping the final version.
Yours sincerely
Jun Li
Reviewer 3 Report
This is a second review after first revisions.
. Authors do not define the site and type of melanoma using ICD-O-3 or ICD-9/ICD10 code.
.Figure 4 has two modes for University of North Carolina. Are these both University of North Carolina, Chapel Hill? It is highly likely.
.Figure 7 does not capitalize the author names.
There is no proper English capitalization for a person's name, country, or university in figures after requesting correction.
Author Response
Dear esteemed reviewer:
We would like to express our heartfelt gratitude for your invaluable feedback and for accepting our revised manuscript. Your expertise and meticulous attention to detail have undoubtedly enhanced the quality and impact of our research. Moreover, we are grateful for your patience and support throughout the review process. Your commitment to maintaining scholarly standards and your willingness to engage in a dialogue with us have been truly commendable. Your guidance has been invaluable in shaping our research and improving its overall coherence. The specific modifications are as follows.
Questions 1:“Authors do not define the site and type of melanoma using ICD-O-3 or ICD-9/ICD10 code.”
Our answer:
We genuinely appreciate your keen observation and bringing this important point to our attention. All the cases included in this article were identified based on the International Classification of Diseases / for Oncology (ICD/ICD-O) to ensure accurate and comprehensive inclusion of related cases, which could help us to reduce clinical heterogeneity.
The WHO international classification of diseases codes (ICDs) is a comprehensive and hierarchical system that assigns unique codes to various diseases, injuries, and health conditions. It serves as a vital tool for health professionals, researchers, and policymakers, facilitating accurate and consistent classification, documentation, and analysis of health data. The transition from ICD-9 to ICD-10 and the recent introduction of ICD-11 have brought improvements in diagnostic accuracy (disease classification), data analysis (coding and documentation), and international comparability (epidemiological research). The application of ICD in melanoma is essential for effective healthcare management, epidemiological studies, and research. Accurate and standardized coding ensures reliable data collection, facilitates international comparability, and enables comprehensive analysis of melanoma trends and outcomes. The continued evolution of ICD reflects the ongoing advancements in medical knowledge and contributes to improving patient care and addressing the global burden of melanoma.
In ICD-9, melanoma is primarily classified under category 172, "Malignant melanoma of skin." This category further includes subcategories based on the anatomic site of the melanoma, such as 172.0 (Face), 172.1 (Ear and external auricular canal), 172.2 (Other and unspecified parts of face), and so on. These codes provide a basic level of specificity regarding the location of the melanoma but lack the necessary granularity to capture additional important clinical details. ICD-9 was widely used until the implementation of ICD-10. ICD-10 introduced significant improvements in coding melanoma. It expanded the coding structure and incorporated additional information for enhanced precision. In ICD-10, melanoma is classified under category C43, "Malignant melanoma of skin." This category is further subdivided based on the anatomical site (C43.0-C43.9) and includes specific codes for melanoma of the eyelid and perianal skin. ICD-10 also includes codes to capture the histopathological types of melanomas, such as superficial spreading melanoma (C43.30), nodular melanoma (C43.31), lentigo malignant melanoma (C43.32), and others. The introduction of ICD-10 has allowed for more accurate recording and monitoring of melanoma cases, facilitating detailed epidemiological analyses and improved patient care. The expanded code set enables capturing variations in tumor behavior, anatomical sites, and histopathological characteristics. This information is vital for healthcare providers, researchers, and policymakers to understand the disease patterns, evaluate treatment outcomes, and develop appropriate interventions. ICD-11 represents the latest advancement in disease classification. It incorporates numerous updates to improve clinical utility and adapt to evolving healthcare practices. In ICD-11, melanoma is classified under category 2C30, "Malignant melanoma of skin." The classification includes detailed codes to specify the anatomical site, histopathological type, and other relevant characteristics of melanoma. One significant enhancement in ICD-11 is the addition of staging information for melanoma. The TNM (Tumor, Node, Metastasis) staging system, widely used in oncology, has been integrated into ICD-11. This allows for capturing critical information regarding tumor size, lymph node involvement, and metastasis. The inclusion of staging information aids in treatment planning, prognosis assessment, and monitoring of disease progression. Furthermore, ICD-11 introduces the concept of extension codes, known as "properties," which provide additional details about the disease. For melanoma, properties include information about ulceration, mitotic rate, regression, and sentinel lymph node involvement. These properties enable a more comprehensive and nuanced description of melanoma cases, facilitating accurate diagnosis, treatment decision-making, and research collaborations.
In summary, ICD/ICD-O provides a valuable tool for the classification, subtyping, and coding of different types and characteristics of melanocytic lesions. It enables the identification of specific subtypes, facilitates data analysis for research purposes, and enhances the comparability and exchange of information across different healthcare institutions and regions. Furthermore, ICD coding captures additional characteristics of melanoma. These parameters help in staging the disease according to widely used systems like the American Joint Committee on Cancer (AJCC) staging system. The precise staging of melanoma assists in determining appropriate treatment strategies, evaluating disease progression, and predicting patient survival. Due to melanoma’s great histological and clinical variability from case to case, ICD/ICD-O help us better classification and clinical management of individual cases in the era of personalized medicine, especially with targeted and personalized vaccine therapy.
Question 2: “Figure 4 has two modes for University of North Carolina. Are these both University of North Carolina, Chapel Hill? It is highly likely.”
Our answer:
Sure, "Univ North Carolina Chapel Hill" and "Uni N Carolina" are the same institution. The first name is the full and official name of the university, while the second name is a shortened and abbreviated version.
We wholeheartedly acknowledge and accept full responsibility for our oversight. It was indeed our negligence to treat different abbreviated names of the same institution as separate entities. We deeply regret the confusion caused by this error and assure you that we will rectify it promptly by making the necessary corrections and redrawing the figure. The revised details is shown in Figure 4 after modification.
We genuinely appreciate your keen observation and bringing this issue to our attention. Your diligence and attention to detail have been instrumental in identifying this mistake, and we are grateful for your meticulous review. Please accept our deepest apologies for any inconvenience caused, and we are committed to addressing this issue promptly and ensuring that our research meets the highest standards of excellence.
Question 3: “Figure 7 does not capitalize the author names.” And “There is no proper English capitalization for a person's name, country, or university in figures after requesting correction.”
Our answer:
We humbly extend our sincere apologies for any confusion that may have arisen. We failed to adequately clarify the underlying issue concerning the representation of names, countries, and institutions in lowercase letters due to a limitation in the software system configuration. We deeply regret any perplexity this may have caused and genuinely appreciate your patience and understanding. The decision to present names, countries, and institutions in lowercase was not intentional but rather a constraint imposed by the software system. Despite our best efforts, we were unable to modify this particular aspect. We fully understand that this may have appeared as an oversight on our part, and we sincerely apologize for not providing a more comprehensive explanation in our previous response.
We acknowledge the importance of upholding academic standards and ensuring clarity in scholarly communication. We deeply regret any unintended confusion that may have resulted from this limitation. It was never our intention to disregard proper capitalization or to compromise the integrity of our work.
Your discerning and meticulous review has been of immense value to us. We greatly appreciate your commitment to ensuring the accuracy and rigor of our research. Your insightful comments and thoughtful suggestions have undoubtedly contributed to the refinement of our manuscript, and we are genuinely grateful for your expertise.
Once again, we offer our sincerest apologies for any misunderstanding or inconvenience caused. Thank you for your understanding, and we assure you that we will make every effort to improve the clarity and presentation of our research in future works.
Yours sincerely
Jun Li
Reviewer 4 Report
I am happy with the responses of the authors.
I recommend this Manuscript for Publication.
Author Response
Dear Esteemed Reviewer,
We would like to express our heartfelt gratitude for your invaluable feedback and for accepting our revised manuscript. Your expertise and meticulous attention to detail have undoubtedly enhanced the quality and impact of our research.
Your insightful comments and constructive suggestions have significantly contributed to refining our work. We greatly appreciate the time and effort you dedicated to thoroughly reviewing our manuscript. Your expertise and thoughtful analysis have not only strengthened the validity of our findings but have also provided us with new perspectives to consider.
We are truly impressed by your profound understanding of the subject matter and the clarity with which you communicated your observations. Your expertise in the field is evident through the depth of your analysis and the comprehensive nature of your comments.
Moreover, we are grateful for your patience and support throughout the review process. Your commitment to maintaining scholarly standards and your willingness to engage in a dialogue with us have been truly commendable. Your guidance has been invaluable in shaping our research and improving its overall coherence.
Once again, we express our sincerest appreciation for your expertise, time, and effort. Your invaluable contributions have undoubtedly made our manuscript stronger and more impactful. We feel privileged to have had the opportunity to benefit from your expertise and insights.
With the utmost gratitude and admiration,
Yours sincerely
Jun Li